

# ISWFoam: A numerical model for internal solitary wave simulation in continuously stratified fluids

Jingyuan Li[1], Qinghe Zhang[1], Tongqing Chen[1]

[1]State Key Laboratory of Hydraulic Engineering Simulation and Safety, Tianjin University, Tianjin 300072, China

*Correspondence to*: Qinghe Zhang (qhzhang@tju.edu.cn)

**Abstract.** A numerical model, ISWFoam, for simulating internal solitary waves (ISWs) in continuously stratified, incompressible, viscous fluids is developed based on a fully three-dimensional (3D) Navier-Stokes equation using the open source code OpenFOAM. This model combines the density transport equation with the Reynolds-averaged Navier-Stokes equation with the Coriolis force, and the model discrete equation adopts the finite volume method. The $k$-$\omega$ SST turbulence model has also been modified accordingly to the variable density field. ISWFoam provides two initial wave generation methods to generate an ISW in continuously stratified fluids, including solving the weakly nonlinear models of the extended Korteweg–de Vries (eKdV) equation and the fully nonlinear models of the Dubreil-Jacotin-Long (DJL) equation. Grid independence tests for ISWFoam are performed, considering the accuracy and computing efficiency, the appropriate grid size of the ISW simulation is recommended to be one-one hundred and fiftieth of the characteristic length and one-twenty fifth of the ISW amplitude. Model verifications are conducted through comparisons between the simulated and experimental data for ISW propagation examples over a flat bottom section, including laboratory scale and actual ocean scale, a submerged triangular ridge, a Gaussian ridge and slope. The laboratory test results, including the ISW profile, wave breaking location, ISW arrival time, and the spatial and temporal changes in the mixture region, are well reproduced by ISWFoam. The ISWFoam model with unstructured grids and local mesh refinement can accurately simulate the generation and evolution of ISWs, the ISW breaking phenomenon and the interaction between ISWs and complex structures and topography.

**Key words.** OpenFoam, Internal solitary wave tank, Stratified fluid, the DJL equation, Grid independence.

## 1. Introduction

Internal solitary waves (ISWs) are commonly observed in oceans, particularly on continental shelf



regions, due to strong tidal current flows over large topographic features (Huthnance, 1981), such as in
the northern South China Sea (Alford et al., 2010; Alford et al., 2015; Cai et al., 2012). ISWs play an
important role in both conveying nutrients from the deep ocean to shallower layers and promoting
biological growth (Sandstrom et al., 1984). Additionally, ISWs are a potential threat to the ocean
structures of resource exploration, exploitation, and submarine navigation vehicles (Alford et al., 2010;
Osborne et al., 1980). A considerable number of studies, which include field measurements, remote
sensing, experiments, theoretical analysis and numerical simulations, have been carried out due to the
significance of ISWs (Vlasenko et al., 2005; Apel et al., 2006; Alford et al., 2011; Guo et al., 2014).
For numerically simulated ISWs, many models have been adopted, including the Euler equation,
the inviscid/viscid incompressible Boussinesq model, the hydrostatic model, the non-hydrostatic model,
and the VOF based two-phase flow model. Among these models, the representative hydrostatic models
include the Naval Research Laboratory Ocean Nowcast/Forecast System (ONFS) (Ko et al.,2008), the
Regional Hallberg Isopycnal Tide Mode (RHIMT) (Hallberg and Rhines, 1996; Hallberg, 1997), and the
Ostrovsky-Hunter model. The representative non-hydrostatic models include the Bergen Ocean Model
(BOM), the nonhydrostatic Regional Ocean Modeling System model (ROMS), the Stanford
Unstructured Nonhydrostatic Terrain-following Adaptive Navier-Stokes Simulator (SUNTANS), and the
Massachusetts Institute of Technology general circulation model (MITgcm). For example, Zhang et al
(2012) established a variable water depth internal wave numerical model in a continuously stratified fluid
system based on the Euler equation. Xu and Stastna (2020) used the viscid incompressible Boussinesq
model to study cross-boundary-layer transport (Boegman and Stastna, 2019) by the fissioning process of
shoaling ISWs. Lamb (1994) established a non-hydrostatic model, using a second-order projection
method developed by Bell and Marcus (1992), which is used for internal wave research including
boundary layer instability (Aghsaee et al., 2012), reflection (Lamb, 2009), and the interaction of the tides
with the topography (Lamb, 2007; Aghsaee et al., 2010). Diamessis (2005) developed a spectral
multidomain penalty method model and correctly reproduced the characteristic vorticity and internal
wave structure. Subich et al (2013) developed a spectral collocation method for the solution of the
Navier–Stokes equations under the Boussinesq approximation, and simulated the internal wave in
continuously stratified fluid. Smedstad et al (2003) employed the ONFS model to establish a global ocean
real-time forecasting system with an operational eddy resolution of 1/16°, which effectively tracks ocean





eddies, ocean currents and ocean fronts. Simmons et al (2004) employed the RHIMT model to carry out
a global numerical simulation of tidal currents, and analyzed the whole process of the conversion rate of
barotropic waves into baroclinic waves. Thiem (2011) used the Bergen Ocean Model to explore the
bottom boundary layer flow caused by waves beneath a propagating ISW in a two-fluid system. Li and
Farmer (2011) employed the Ostrovsky-Hunter model to study the nonlinear evolution of a
monochromatic internal wave. Buijsman et al (2010) employed ROMS model to study the asymmetry in
solitons to the east and west of Luzon Strait. Zhang et al (2011) used the nonhydrostatic SUNTANS
model (Fringer et al., 2006) to study the dynamics of A wave and B wave formation. Rayson et al (2018)
used the modified SUNTANS model to study the internal waves around Scott Reef and provided the
generation process of internal lee waves. Vlasenko et al (2010) employed the MITgcm model to
investigate the baroclinic tidal energy conversion in the area west of the Luzon Strait.
In summary, for continuously stratified fluids in complex ocean environments, numerical simulation
has become a leading method for ISW investigations. However, there are presently few versatile
numerical models with share code that can accurately simulate the ISW flow around complex topography
and submarine navigation vehicles in continuously stratified fluids. Therefore, the main objective of this
paper is to develop a solver, referred to as ISWFoam with a modified *k-ω SST* model that considers the
variable density field, which simulates the ISW in continuous density stratification, incompressible and
viscous fluids using the finite volume method with unstructured grids based on a fully three-dimensional
(3D) Navier-Stokes equation using the OpenFOAM library.
Notably, the open source field operation and manipulation code OpenFOAM®, as an object-
oriented C++ open source library that can be used to build a variety of solvers for computational fluid
problems based on the finite volume method, is becoming increasingly popular in the computational fluid
research community. At present, the official version of OpenFOAM® does not have a solver or boundary
conditions for solving the ISW in continuously stratified fluids. Although some researchers simulate
ISWs by modifying the OpenFOAM® code, most of these studies are based on a two-fluid system
without considering continuous stratification in density, such as Meng and Zhang (2016) and Li et al
(2017). Though recent work by Ding et al (2020) and Li et al (2021) considered continuous stratification
in density, the wave generation method is essential for a two-layer system. To extensively use of the
numerical model of ISWs as a tool in the future, we will develop ISWFoam to simulate the ISW in





continuously stratified, incompressible and viscous fluids based on the OpenFOAM library. The
turbulence model will consider the variable density field. In addition, ISWFoam will provide two initial
methods to generate an ISW in continuously stratified fluids, including solving the weakly nonlinear
models of the extended Korteweg-de Vries (eKdV) equation and the fully nonlinear models of the
Dubreil-Jacotin-Long (DJL) equation. This approach renders the numerical model suitable for the
simulation of ISW flows in complex geometries and topographies.
The outline of the paper is described as follows. First, in Section 2, the governing equations for a
continuously stratified fluid are presented, and discrete forms of these equations are derived. Then, grid
independence tests of the developed ISWFoam model are described in Section 3. Subsequently, in
Section 4, a series of test cases are presented to verify the model. Finally, the conclusions are drawn in
Section 5.
**2. ISWFoam: A three-dimensional numerical solver for ISWs in a continuously stratified fluid**
**2.1 Governing equations**
We present an ISW numerical model by solving the motion of a three-dimensional, viscous,
incompressible fluid with the Boussinesq approximation and rigid lid hypothesis. The governing
equations of the model are
$$\nabla \cdot \mathbf{U} = 0, \tag{1}$$
$$\frac{\partial \mathbf{U}}{\partial t} + (\mathbf{U} \cdot \nabla)\mathbf{U} - \nabla \cdot (\nu_{Eff}\nabla\mathbf{U}) = \mathbf{Q} \quad \mathbf{Q} = \frac{1}{\rho_0}\left(-\nabla p_{\_rgh} - \mathbf{g}\cdot\mathbf{X}\nabla\rho - \Omega e_3\right), \tag{2}$$
$$\frac{\partial \rho}{\partial t} + (\mathbf{U} \cdot \nabla)\rho = \nabla \cdot (k\nabla\rho), \tag{3}$$
where $\mathbf{U} = (u_i, u_j, u_k)$ is the velocity vector, $t$ is time, $\nabla$ is the gradient operator, $\mathbf{Q}$ is the source term,
$\rho_0$ is the reference density, $\rho$ is the density field, $p_{\_rgh}$ is a modified pressure field, $\mathbf{g}$ is the gravitational
acceleration vector, and $\mathbf{X}$ is the position vector. $\nu_{Eff}$ is the effective kinematic viscosity defined as $\nu_{Eff} =$
$\mu_{Eff}/\rho_0$, where $\mu_{Eff}$ is the effective dynamic viscosity including the molecular viscosity ($\mu_l$) and turbulent
viscosity($\mu_t$). $k$ is the diffusion coefficient, and its value is the same as the effective dynamic
viscosity($\mu_{Eff}$). $\Omega$ is the Coriolis parameter, which is the twice the speed of rotation around the vertical
unit vector $e_3 = (0,0,1)$. ISWFoam uses a modified pressure $p_{\_rgh}$ instead of a total pressure $p$, and their
relationship is given by





$$p_{\_rgh}=p-\rho\mathbf{g}\cdot\mathbf{X},\ \nabla p_{\_rgh}=\nabla p-\rho\mathbf{g}-\mathbf{g}\cdot\mathbf{X}\nabla\rho,$$ (4)
To close the above equations, the turbulence model needs to be employed. The two-equation $k$-$\varepsilon$
model is widely used as an effective turbulence model, but it cannot capture the proper behaviour of
turbulent boundary layers up to separation due to adverse pressure gradients (Wilcox, 1993). For the
above boundary layers separation problem, Bardina et al. (1997) and Menter et al. (2003) suggested the
use of the $k$-$\omega$ $SST$ model to obtain substantially more accurate results. Therefore, the turbulence model
used in this paper is the $k$-$\omega$ $SST$ model. Notably that in OpenFOAM, the incompressible version for
turbulence models does not consider the variable density field, and instead, it treats the density as a
constant, such as the $k$-$\omega$ $SST$ model
$$\frac{\partial k}{\partial t}+\nabla\cdot(\mathbf{U}k)=\nabla\cdot\left[\left(\nu_{Eff}+\sigma_{k}\nu_{t}\right)\nabla k\right]+P_{k}^{*}-\beta^{*}\omega k$$ (5)
$$\frac{\partial\omega}{\partial t}+\nabla\cdot(\mathbf{U}\omega)=\nabla\cdot\left[\left(\nu_{Eff}+\sigma_{\omega}\nu_{t}\right)\nabla\omega\right]+C_{\gamma}\frac{\omega}{k}P_{k}-C_{\beta}\omega^{2}+2\left(1-F_{1}\right)\frac{\sigma_{\omega2}}{\omega}\nabla k\cdot\nabla\omega$$ (6)
$$P_{k}^{*}=\min(P_{k},c_{1}C_{\mu}k\omega)$$ (7)
$$\nu_{t}=\frac{a_{1}k}{\max\left(a_{1}\omega,\sqrt{2}S_{t}F_{2}\right)}$$ (8)
where $k$ is the turbulent kinetic energy, $\omega$ is the specific dissipation rate, $P_{k}$ is the production term of $k$,
$P_{k}=\tau^{R}:\nabla\mathbf{U}$, $P_{k}^{*}$ is related to the production term of turbulence kinetic energy $P_{k}$ in the $k$ equation, $\nu_{t}$
is the turbulent kinematic viscosity, $S_{t}$ is the mean rate of the flow strain, $S_{t}=0.5(\nabla\mathbf{U}+\nabla\mathbf{U}^{T})$, the model
constants are assigned the values $\beta^{*}=0.09$, $a_{1}=0.31$, $c_{1}=10$ and $C_{\mu}=0.09$, $F_{1}$ and $F_{2}$ are blending
functions, the value of $\sigma_{k}$, $\sigma_{\omega}$, $C_{\gamma}$ and $C_{\beta}$ are blended using the equation $\Phi=F_{1}\Phi_{1}+(1-F_{1})\Phi_{2}$ in which
$\Phi_{1}$ and $\Phi_{2}$ are given in Table 1.

133                          Table 1 Default values for $\Phi_{1}$ and $\Phi_{2}$

| $\Phi$ | $\sigma_{k}$ | $\sigma_{\omega}$ | $C_{\beta}$ | $C_{\gamma}$ |
|---|---|---|---|---|
| $\Phi_{1}$ | 0.85 | 0.5 | 0.075 | 5/9 |
| $\Phi_{2}$ | 1.0 | 0.856 | 0.0828 | 0.44 |


Considering the variable density field during the solution process, it is necessary to consider the
change in the density field in the turbulence model. Therefore, we modify the turbulence model to
consider the change in density, and finally a modified $k$-$\omega$ $SST$ model that considers the change in density
is used to close the equation



$$\frac{\partial \rho k}{\partial t} + \nabla \cdot (\rho \mathbf{U} k) = \nabla \cdot \left[ \rho \left( \nu_{Eff} + \sigma_k \nu_t \right) \nabla k \right] + \rho P_k^* - \rho \beta^* \omega k \qquad (9)$$
$$\frac{\partial \rho \omega}{\partial t} + \nabla \cdot (\rho \mathbf{U} \omega) = \nabla \cdot \left[ \rho \left( \nu_{Eff} + \sigma_\omega \nu_t \right) \nabla \omega \right] + C_\gamma \frac{\omega}{k} P_k - C_\beta \rho \omega^2 + 2 \left( 1 - F_1 \right) \rho \frac{\sigma_{\omega 2}}{\omega} \nabla k \cdot \nabla \omega \qquad (10)$$

**2.2 Numerical discretization**

The governing equations are numerically discretized using the finite volume method based on the

C++ open source library of OpenFOAM. The finite volume method requires that Eqs. (2) and (3) are
satisfied over the control volume $V_P$ around point P in integral form:
$$\int_{V_P} \int_{\Delta t} \left[ \frac{\partial \mathbf{U}}{\partial t} + (\mathbf{U} \cdot \nabla) \mathbf{U} - \nabla \cdot (\nu_{Eff} \nabla \mathbf{U}) \right] dV dt = \int_{V_P} \int_{\Delta t} \mathbf{Q} \, dV dt, \qquad (11)$$
$$\int_{V_P} \int_{\Delta t} \left[ \frac{\partial \rho}{\partial t} + (\mathbf{U} \cdot \nabla) \rho - \nabla \cdot (k \nabla \rho) \right] dV dt = 0, \qquad (12)$$

The momentum equation in ISWFoam is solved by constructing a predicted velocity field and then

using the Pressure Implicit with Splitting of Operators (PISO) algorithm (Issa, 1986) to modify it. *n* is
defined to represent the current moment. The PISO iteration process is marked as *m*; when *m* is equal to
zero, it represents the initial moment ($t^n$).

First, only the temporal, convection and diffusion terms appear in the discrete version of the

equation momentum, and the other terms are ignored. After this operation, we obtain an explicit
expression for the predicted velocity field $\mathbf{U}_P^r$, namely,
$$\frac{\mathbf{U}_P^r - \mathbf{U}_P^n}{\Delta t} V_P + \sum_{f \in \partial V_P} \left( \phi_f^n \mathbf{U}_f^r \right) - \sum_{f \in \partial V_P} \nu_{Eff} \nabla \mathbf{U}_f^r \cdot \mathbf{S}_f = 0, \qquad (13)$$
where *P* represents the centre of the grid cell, $\phi_f^n = \mathbf{U}_f^n \cdot \mathbf{S}_f$ is the volume flux at the initial time *n* and
$\mathbf{S}_f$ is the face vector.

The solution process requires the velocity on the surface *f*. Assuming the variation in $\mathbf{U}_f^r$ between

the centre *P* of the grid and the centre *N* of the adjacent grid, the face values are calculated using a mixture
method (blended differencing) of the central scheme (central differencing) and the upwind scheme
(upwind differencing) as follows (Jasak, 1996):
$$\mathbf{U}_f = \left( 1 - \lambda_U \right) \left( \mathbf{U}_f \right)_{UD} + \lambda_U \left( \mathbf{U}_f \right)_{CD} \qquad (14)$$
where





$$\left(\mathbf{U}_f\right)_{UD} = \begin{cases} \mathbf{U}_P \text{ for } \phi_f \geq 0, \\ \mathbf{U}_N \text{ for } \phi_f < 0, \end{cases} \quad \text{and} \quad \left(\mathbf{U}_f\right)_{CD} = \frac{\mathbf{U}_P + \mathbf{U}_N}{2} \tag{15}$$

where $N$ represents the centre of the adjacent grid cells, $\phi_f = \mathbf{U}_f \cdot \mathbf{S}_f$ is volume flux. The limiter $\lambda_U$
can be selected from several alternatives (OpenFOAM, 2019), including linear, QUICK, vanLeer, etc. In
the following derivation process, the vanLeer scheme was used to calculate the velocity of the face centre
$$\mathbf{U}_f = \frac{1}{2}\left(\mathbf{U}_P + \mathbf{U}_N\right) + \frac{1}{2}\left[\psi(\phi_f)(1-\lambda_U)\right]\left(\mathbf{U}_P - \mathbf{U}_N\right), \tag{16}$$

where $\psi(\phi_f)$ is a step function defined by
$$\psi(\phi_f) = \begin{cases} 1 \text{ for } \phi_f \geq 0, \\ -1 \text{ for } \phi_f < 0, \end{cases} \tag{17}$$

Inserting Eq. (16) into Eq. (13) yields
$$A_P \mathbf{U}_P^r = \sum_{f \in \partial V_P} A_N \mathbf{U}_P^m + \frac{\mathbf{U}_P^n}{\Delta t} = H(\mathbf{U}^m) \tag{18}$$

After some manipulation, the quantities $A_P$ and $A_N$ are given as
$$A_P = \left\{ \frac{V_P}{\Delta t} + \sum_{f \in \partial V_P} \frac{\phi_f^n}{2}\left[1 + \psi(\phi_f)(1-\lambda_U)\right] + \sum_{f \in \partial V_P} v_{Eff,f} \frac{|\mathbf{S}_f|}{|d|} \right\} \frac{1}{V_P} \tag{19}$$

$$A_N = \left\{ -\frac{\phi_f^n}{2}\left[1 - \psi(\phi_f)(1-\lambda_U)\right] + v_{Eff,f} \frac{|\mathbf{S}_f|}{|d|} \right\} \frac{1}{V_P} \tag{20}$$

Including the effect of gravity and the Coriolis force in Eq. (18)
$$\mathbf{U}_P^r = \frac{H(\mathbf{U}^m)}{A_P} - \frac{\left(\mathbf{g} \cdot \mathbf{X} \nabla \rho / \rho_0\right)^n}{A_P} - \frac{\left(\Omega e_3\right)^n}{A_P}, \tag{21}$$

Notably, that when $m$ is equal to zero, it represents the initial moment $n$, and the value of the initial
moment is known. Therefore, we obtain the predicted velocity field $\mathbf{U}_P^r$ in the first iteration. We define
the surface gradient operator ( $\nabla \frac{1}{f}$ ), and the type of gradient operator acting on $\mathbf{U}$ is
$\nabla \frac{1}{f} \mathbf{U} = \left(\mathbf{U}_N^m - \mathbf{U}_P^m\right)\big/|d|$, which represents the distance from the centre of the grid $N$ to $P$. Similarly,
the surface gradient operator ( $\nabla \frac{1}{f}$ ) acting on scalar $\gamma$ is $\nabla \frac{1}{f} \lambda = \left(\lambda_N^m - \lambda_P^m\right)\big/|d|$. The associated flux
( $\phi_f = \mathbf{U}_f \cdot \mathbf{S}_f$ ) is achieved by executing an inner product with a surface vector ($\mathbf{S}_f$) on the left and right
parts of Eq. (21), giving



$$\phi_f^r = \left(\frac{H(\mathbf{U}^m)}{A_P}\right)_f \cdot \mathbf{S}_f - \left(\left(\frac{1}{A_P}\right)_f (\mathbf{g}\cdot\mathbf{X})_f^n \left(\frac{1}{\rho_0}\nabla_{\frac{1}{f}}\rho\right)^n |\mathbf{S}_f|\right) - \left(\frac{(\Omega e_3)^n}{A_P}\right)_f \cdot \mathbf{S}_f, \tag{22}$$

Eq. (22) completed the flux calculation without considering the influence of the pressure term. The
pressure contribution in terms of a flux can be expressed as
$$\left(\frac{-\nabla p_{\_rgh}}{\rho_0 A_P}\right)_f \cdot \mathbf{S}_f = \left(\frac{-1}{A_P}\right)_f \left(\frac{1}{\rho_0}\nabla_{\frac{1}{f}} p_{\_rgh}^{m+1}\right)|\mathbf{S}_f|, \tag{23}$$

Then, Eq. (23) is now added to Eq. (22) to yield
$$\phi_f^{m+1} = \left(\frac{H(\mathbf{U}^m)}{A_P}\right)_f \cdot \mathbf{S}_f - \left(\left(\frac{1}{A_P}\right)_f (\mathbf{g}\cdot\mathbf{X})_f^n \left(\frac{1}{\rho_0}\nabla_{\frac{1}{f}}\rho\right)^n |\mathbf{S}_f|\right) - \left(\frac{(\Omega e_3)^n}{A_P}\right)_f \cdot \mathbf{S}_f - \left(\frac{1}{A_P}\right)_f \left(\frac{1}{\rho_0}\nabla_{\frac{1}{f}} p_{\_rgh}^{m+1}\right)|\mathbf{S}_f| \tag{24}$$

Combined with Eq. (22), Eq. (24) is simplified and rewritten as
$$\phi_f^{m+1} = \phi_f^r - \left(\frac{1}{A_P}\right)_f \left(\frac{1}{\rho_0}\nabla_{\frac{1}{f}} p_{\_rgh}^{m+1}\right)|\mathbf{S}_f| \tag{25}$$

Using conservation of mass, we solve the pressure field $p_{\_rgh}^{m+1}$, which results in
$$\sum_{f\in\partial V_P} \left(\frac{1}{A_P}\right)_f \left(\frac{1}{\rho_0}\nabla_{\frac{1}{f}} p_{\_rgh}^{m+1}\right)|\mathbf{S}_f| = \sum_{f\in\partial V_P} \phi_f^r \tag{26}$$

The preconditioned conjugate gradient method is used to solve the linear system constructed by Eq.

(26) (OpenFOAM, 2019). After $p_{\_rgh}^{m+1}$ is obtained using Eq. (26), we calculate the volume flux using
Eq. (25) for each face. The cell centred velocity fields $\mathbf{U}_P^{m+1}$ are calculated by reconstructing the face
velocity flux using the following expression (Deshpande, 2012)
$$\mathbf{U}_P^{m+1} = \mathbf{U}_P^r + \left(\frac{1}{A_P}\right)\left(\sum_{f\in\partial V_P}\frac{\mathbf{S}_f\mathbf{S}_f}{|\mathbf{S}_f|}\right)^{-1} \cdot \left(\sum_{f\in\partial V_P}\left(\frac{\phi_f^{m+1} - (\mathbf{U}_P^r)_f \cdot \mathbf{S}_f}{(1/A_P)_f}\right)\frac{\mathbf{S}_f}{|\mathbf{S}_f|}\right) \tag{27}$$

Eq. (27) completes the velocity field calculation of the first iteration step in the PISO algorithm. By

converting the identifier $m$ to $m+1$, the next PISO iteration is completed and updating the velocity in Eq.
(18) with the velocity $\mathbf{U}_P^{m+1}$ calculated from Eq. (27), thereby updating $p_{\_rgh}$, $\phi_f$ and $\mathbf{U}$. This procedure
is performed $M$ times to guarantee that the results of the velocity and pressure together conform to the
continuity and momentum equations. Considering that PISO iteration levels require more than 1, but
typically not more than 4 (OpenFOAM, 2019), we specify that the number of PISO iteration levels is 3



in the computations presented in this paper. After completing the three iterations, the converged values
are considered the result of the next time step ($n + 1$), namely,
$$\phi_f^{n+1} = \phi_f^M, \quad \mathbf{U}_P^{n+1} = \mathbf{U}_P^M, \quad p_{\_rgh}^{n+1} = p_{\_rgh}^M,$$ (28)
We discretize the convection-diffusion equation of density (12) to obtain
$$\frac{V_P}{\Delta t}(\rho_P^{n+1} - \rho_P^n) + \sum\left(\phi_f^{n+1}\rho_f^{n+1}\right) = \sum k\left[|\mathbf{S}_f|\frac{\rho_N^{n+1} - \rho_P^{n+1}}{|d|}\right],$$ (29)
At the end of the iteration procedure, we bring the results of the volume flux into Eq. (29) to
calculate the density field at the next time ($\rho_P^{n+1}$), thereby updating the density field for the next step
calculation ($\Delta t = t^{n+2} - t^{n+1}$).
**2.3 Initialized field of ISW generation**
ISW generation methods mainly include the gravity collapse mechanism, double push-pedals
method (Fu et al., 2008), velocity-inlet method (Gao et al., 2012), mass source method (Wang et al.,
2018), initialization method, and methods addressing the interaction between tidal current and
topography. For example, Hsieh et al (2014) investigated the flow evolution of a depression ISW
generated by the gravity collapse mechanism. Cheng et al (2020) studied the interaction between ISWs
and a cylinder using the gravity collapse mechanism. The initialization method involves solving the
internal solitary wave theory at the initial moment, such as the Korteweg-de Vries (KdV) equation
(Grimshaw et al., 2010), the modified KdV (mKdV) equation, the extended KdV (eKdV) equation, the
forced KdV equation, the Ostrovsky equation (Li and Farmer, 2011), the Miyata-Choi-Camassa (MCC)
model (Miyata 1985 and 1988; Choi and Camassa, 1999), and the Dubreil-Jacotin-Long (DJL) equation
(Long, 1953; Turkington, 1991; Brown and Christie, 1997; Dunphy et al., 2011), to obtain the wave
surface, velocity field. The method of an interacting between tidal current and terrain that stimulates
ISWs is adopted by many scholars, such as Farmer and Smith (1980), Lamb et al (1994), and Shaw et al

(2009).

In this paper, the method of initializing the field is selected to generate the ISWs. To increase the
application range of the ISWFoam model, two initialization methods are provided, including solving the
weakly nonlinear models of the eKdV equation (Helfrich and Melville, 2006) and the fully nonlinear
models of the DJL equation for continuously stratified fluids (Turkington, 1991; Dunphy et al, 2011).





The Dubreil-Jacotin-Long (DJL) equation is expressed as

$$\nabla^2 \eta + \frac{N^2(z-\eta)}{c^2}\eta = 0, \quad \eta = 0 \quad \text{at} \quad z = 0, -H$$
$$\eta = 0 \quad \text{at} \quad |x| \to \infty \tag{30}$$

where $\eta$ is the isopycnal displacement, $H$ is the water depth, $c$ is the propagation speed, $N$ is the definition
of the buoyancy frequency, and $z$ is vertical position.

$$N^2(z) = -g\frac{d\rho_0(z)}{dz}, \tag{31}$$

where $\rho_0(z)$ is the reference density, and $g$ is the gravitational acceleration.
By solving the above DJL equation we can obtain $\eta$ and $c$, and then through the relationship $\psi = \eta c$,
where $\psi$ is the stream function, we can obtain the wave-induced velocity field. We use the DJLES open
source package provided by Dunphy et al (2011) to solve the DJL equations. Then we input the initial
field calculated by DJLES into OpenFOAM to obtain the initial field required for OpenFOAM numerical
simulation.
Another theory of ISWFoam model wave generation involves the weakly nonlinear models of the
eKdV equation. Using the first order stream function for the DJL equation, we can obtain the well-known
KdV equation and further obtain the eKdV equation. For the specific derivation, please refer to the paper
by Lamb and Yan (1996). The eKdV equation (Helfrich and Melville, 2006) is

$$\frac{\partial \zeta}{\partial t} + \left(c_0 + c_1\zeta + c_3\zeta^2\right)\frac{\partial \zeta}{\partial x} + c_2\frac{\partial^3 \zeta}{\partial x^3} = 0, \tag{32}$$

$$c_0^2 = \frac{gh_1h_2(\rho_2 - \rho_1)}{\rho_1h_2 + \rho_2h_1}, \tag{33}$$

$$c_1 = -\frac{3c_0}{2}\frac{\rho_1h_2^2 - \rho_2h_1^2}{\rho_1h_1h_2^2 + \rho_2h_1^2h_2}, \tag{34}$$

$$c_2 = \frac{c_0}{6}\frac{\rho_1h_1^2h_2 + \rho_2h_1h_2^2}{\rho_1h_2 + \rho_2h_1}, \tag{35}$$

$$c_3 = \frac{3c_0}{h_1^2h_2^2}\left[\frac{7}{8}\left(\frac{\rho_1h_2^2 - \rho_2h_1^2}{\rho_1h_2 + \rho_2h_1}\right)^2 - \frac{\rho_1h_2^3 + \rho_2h_1^3}{\rho_1h_2 + \rho_2h_1}\right], \tag{36}$$

where $\zeta$ is the isopycnal vertical displacement; $h_1$ and $h_2$ are the mean upper and lower layer depths,

respectively; $\rho_1$ and $\rho_1$ are the fluid densities of the upper and lower layers, respectively. The theoretical





solution of Eq. (32) above is

$$\zeta = \frac{a}{B + (1-B)\cosh^2\left[\lambda_{\text{eKdV}}\left(x - c_{\text{eKdV}}t\right)\right]}, \tag{37}$$

$$\lambda_{\text{eKdV}}^2 = \frac{a}{12c_2}\left(c_1 + \frac{1}{2}c_3 a\right), \tag{38}$$

$$c_{\text{eKdV}} = c_0 + \frac{a}{3}\left(c_1 + \frac{1}{2}c_3 a\right), \tag{39}$$

$$B = \frac{-ac_3}{2c_1 + ac_3}, \tag{40}$$

$$u_1 = -c_{\text{eKdV}}\frac{\zeta}{h_1 - \zeta}, \quad u_2 = c_{\text{eKdV}}\frac{\zeta}{h_2 + \zeta}, \tag{41}$$

where $a$ is the ISW amplitude, $\lambda_{\text{eKdV}}$ is the wavelength, $c_{\text{eKdV}}$ is the wave speed, $B$ is an auxiliary parameter,
and $u_1$ and $u_2$ are the speeds of the upper and lower layers of the fluid, respectively. The waveform and
velocity field of the ISWs are solved at the initial moment by the developed function and then assigned
to the calculation domain.
The vertical profile of the initial density is given by a hyperbolic tangent function profile (Aghsaee
et al., 2010)

$$\overline{\rho}(z) = \frac{\rho_1 + \rho_2}{2} - \frac{\rho_2 - \rho_1}{2}\tanh\left(\frac{z - z_{pyc}}{d_{pyc}}\right) \tag{42}$$

where $z$ is the vertical position; $\rho_1$ and $\rho_2$ are the fluid densities of the upper and lower layers, respectively;
$z_{pyc}$ is the location of the centre of the pycnocline; and $d_{pyc}$ is the thickness of the pycnocline. In this paper,
unless otherwise specified, the form of the density profile adopts Eq. (42). The internal solitary wave
surface is obtained by calculating the gradient of the density field, and the absolute value of the maximum
value of the gradient represents the vertical position of the wave surface. Notably, the density profile of
the actual ocean is not always hyperbolic, so our model provides a function for users to modify the
density profile according to the actual situation.
**2.3.1 Comparison between the DJL equation and the eKdV equation**
To compare the DJL equation and the eKdV equation, we set up a numerical simulation, which
includes a tank that is 15 m long, 1 m wide and has a water depth of 0.5 m. The depths of the upper ($h_1$)
and lower ($h_2$) layers are 0.1 m and 0.4 m, respectively, the densities of the upper and lower layers are
1022 kg/m³ and 1028 kg/m³, respectively, the location of the centre of the pycnocline ($z_{pyc}$) is 0.4 m, the





pycnocline thickness ($d_{pyc}$) is 0.04 m vertically, the initial ISW amplitude ($a$) is 0.065 m and the location
of the centre of ISW is 12.5m. The ISWs propagate from right to left. The measuring point P is set at a
position 10m away from the initial ISW. The grid is uniform in the $x$-direction, $y$-direction and $z$-direction,
and the sizes are $\Delta x = 1 \times 10^{-2}$ m, $\Delta y = 1 \times 10^{-2}$ m and $\Delta z = 1 \times 10^{-3}$ m, respectively. Slip boundary conditions
are applied to the bottom and both sides, while cyclic boundary conditions are assigned to the inlet and
outlet boundaries. The top boundary is a rigid lid. The boundary conditions related to the density field
are no-flux boundary conditions.

Fig. 1 shows the comparison of the horizontal velocity component field when the DJL equation and

the eKdV equation are used to generate ISWs. At the initial moment, the ISW generated by the eKdV
equation is not as smooth as the ISW generated by the DJL equation, and the horizontal velocity at the
interface area is discontinuous as shown in Fig. 1($a$) and ($b$). With the propagation of ISWs, the ISWs
generated by the DJL equation are always smooth at the interface area, and the velocity field is always
continuous as shown in Fig. 1($a$), ($c$), ($e$) and ($g$). Correspondingly, the ISW generated by the eKdV
equation gradually produces a gradient in the vertical direction of the horizontal velocity in the interface
area, thus, the interface area becomes smooth, and the velocity becomes continuous. Fig. 1($d$) shows this
evolution process, which is basically completed in 5s as shown in Fig. 1($f$). At 50s, the difference between
the horizontal velocity fields of the two equations is very small as shown in Fig. 1($g$) and ($h$).

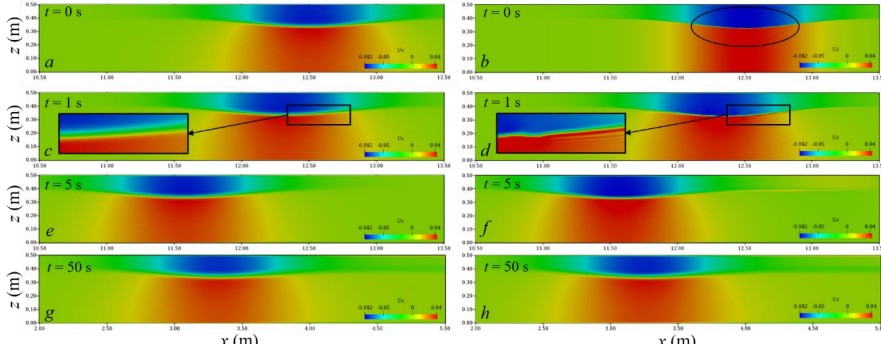


Figure 1: Comparison chart of the horizontal velocity component field: DJL equation (left) and eKdV equation

(right).

Fig. 2 shows the comparison of the vertical velocity component field when the DJL equation and

the eKdV equation are used to generate ISWs. Since the theoretical solution of the eKdV equation only
obtain the average horizontal velocity of the upper and lower layers of the fluid, there is no vertical
velocity at the initial moment, as shown in Fig. 2($b$). With the propagation of ISWs, the vertical velocity
field will gradually be generated and finally stabilized, and the stable time occurs at 5s as shown in Fig.
2(*b*), (*d*), (*f*) and (*h*). At 50s, the difference between the vertical velocity fields of the two equations is
very small as shown in Fig. 2(*g*) and (*h*).

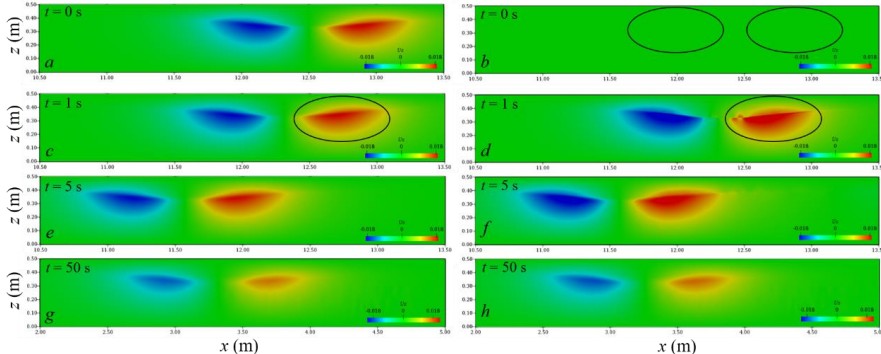


Figuire 2: Comparison chart of the vertical velocity component field: DJL equation (left) and eKdV equation
(right).

The ISW propagates for 10 m, and the amplitudes of the ISWs generated by the DJL equation and

the eKdV equation are reduced by 9.88% and 17.96%, respectively, as shown in Fig. 3. Overall, the
reduction in energy leads to the attenuation of the amplitude of the ISW, which in turn reduces the wave
speed. Except for the difference in initial fields, the grid sizes, time step, turbulence model, and other
features are the same. Therefore, the initial stage of ISWs generated by the eKdV equation leads to
excessive energy loss compared with those generated by the DJL equation. From the above analysis of
the velocity field, we know that the method of initializing the field with the eKdV equation requires a
period of movement before the jump of the velocity field develops into a field with continuous changes
in velocity. In addition, the DJL equation, as a fully nonlinear model, can better reflect its superiority for
internal waves with strong nonlinearity. Therefore, the wave generation of the subsequent numerical
cases in this paper adopts the method of initializing the field with the DJL equation.

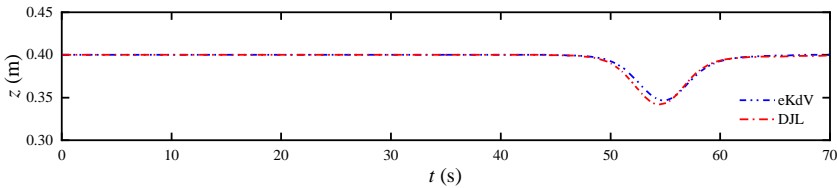


Figure 3: Time series of the interface displacement. The probe was 10 m away from the initial ISW.





**3. Grid independence of the ISW simulation**

These grid independence tests were performed in the horizontal and vertical directions by applying meshes of different sizes. The sizes of the mesh determined in this paper are calculated based on the amplitude of the ISW and a characteristic length determined through the integration of the wave profile (Michallet and Ivey, 1999)

$$L = \frac{1}{a} \int_{-\infty}^{\infty} \zeta(x) dx \qquad (43)$$

where $\zeta$ is the isopycnal vertical displacement and $a$ is the ISW amplitude.

To determine the appropriate mesh size, the propagation of ISWs on flat bottoms is calculated, and the numerical results are compared with the DJL theoretical solution. We set up a numerical simulation, which includes a tank that is 50 m long, 0.5 m wide and has a water depth of 0.5 m. The depths of the upper ($h_1$) and lower ($h_2$) layers are 0.1 m and 0.4 m, respectively, the densities of the upper and lower layers are 1000 kg/m³ and 1030 kg/m³, respectively, the location of the centre of the pycnocline ($z_{pyc}$) is 0.4 m, and the pycnocline thickness ($d_{pyc}$) is 0.05 m vertically, the ISW amplitude ($a$) is 0.065 m. The measuring point P is set at a position 10$L$ away from the initial ISW. The sponge layer on both sides, whose length is the double wave characteristic length, has been checked to properly dissipate the reflected wave. Slip boundary conditions are applied to the bottom and both sides, while cyclic boundary conditions are assigned to the inlet and outlet boundaries. The top boundary is a rigid lid. The boundary conditions related to the density field are no-flux boundary conditions.

**3.1 Grid independence in the horizontal direction**

First, we analyse the grid independence in the horizontal direction, with a constant cell height of $\Delta z = a/20$ m. Fig. 4 shows the results of the comparison of the waveform and decay rate in the horizontal direction at probe P1 with the ISWFoam using a wide range of grid configurations. The results show a negligible difference in the waveform when the mesh size is less than $L/40$, so it is difficult to accurately analyse the grid independence just by the waveform. A traditional decay rate parameter is adopted, namely $\delta = (a_{probe} - a_{initial})/a_{initial}$, where $a_{initial}$ is the ISW amplitude value at the initial moment, $a_{probe}$ is the ISW amplitude value of the probe 10$L$ away from the initial ISW. Fig. 4(b) shows the relationship between the decay rate of the ISW amplitude and the grid quantity per unit length for different mesh sizes. As shown in Fig. 4(b), the decay rate of the ISW amplitude tends to be smooth as the grid number



per unit length increases to 160 ($\Delta x = L/150$), and then the increase in the grid quantity has a relatively
small effect on the decay rate. Therefore, for ISWFoam developed in this paper, we suggest that the
dimensions of the horizontal grid are $L/150$.

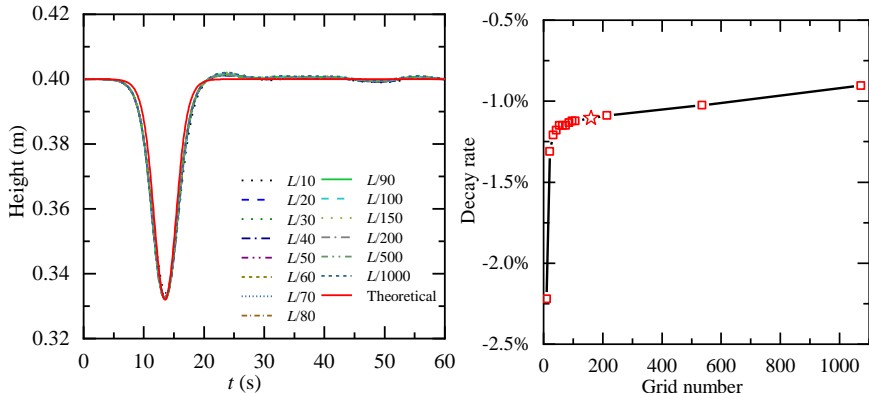

Figure 4: Grid independence in the horizontal direction at probe P1: (a) comparison of waveform and (b) decay
rate.

### 3.2 Grid independence in the vertical direction

Second, we analyse the grid independence in the vertical direction, with a constant cell width of $\Delta x$
$= L/150$ m. Fig. 5 shows the results of a comparison of the waveform and decay rate of the ISW amplitude
in the vertical direction at probe P1 with the ISWFoam using a wide range of grid configurations. The
results also show a negligible difference in the waveform when the mesh size is less than $a/10$, so it is
difficult to accurately analyse the grid independence just by the waveform. As shown in Fig. 5(b), the
decay rate of the ISW amplitude decreases as the grid quantity increases in a wave height range before
the numerical oscillation occurs. Here, we assume that the grid size with the decay rate of the ISW
amplitude less than one percent is the appropriate vertical grid size; namely, the vertical grid size is $a/25$
m. Therefore, for ISWFoam developed in this paper, we suggest that the dimensions of the vertical grid
be $a/25$.



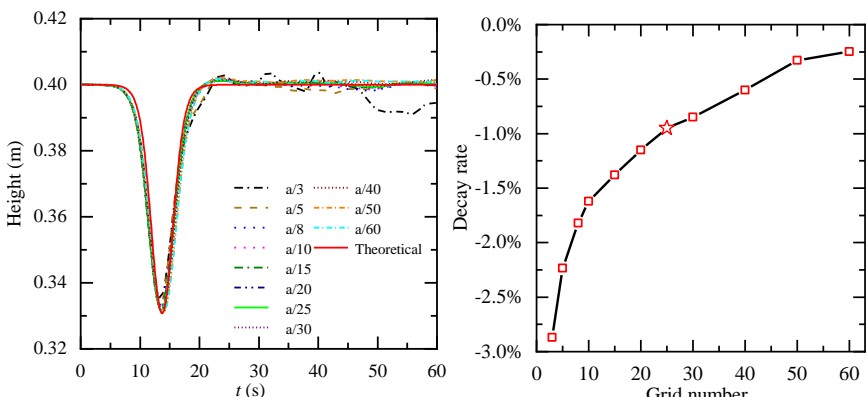

Figure 5: Grid independence analysis in the vertical direction at probe P1: (a) comparison of waveform and (b)
decay rate.
Finally, for ISWFoam developed in this paper, we suggest that the dimensions of the horizontal grid
are $L/150$, while the vertical grid is $a/25$.
**4. Model verification and results**
To verify the numerical model of the ISWs, the propagation of ISWs on a flat bottom section,
submerged triangular ridge and slopes is calculated, and the numerical results are compared with the
corresponding experimental results. To verify the correctness of Coriolis code implantation and reflect
the role of local mesh refinement, the propagation of ISWs on a flat bottom section of actual ocean scale
and a Gaussian ridge is calculated.
**4.1 ISW propagating on a flat bottom section**
**4.1.1 Experimental data used**
In this section, ISWFoam is verified by employing ISWs propagating on a flat bottom section with
Case Flat_4 in the continuously stratified laboratory experiment described in Hsieh et al. (2014). The
physical dimensions and ultrasonic probe locations in the experiments of Hsieh et al. (2014), as shown
in Fig. 6, are adopted to establish the numerical computation domain. We set up a numerical tank of the
experiment of Hsieh and co-authors, which includes a tank that is 15 m long, 0.5 m wide and has a stable
water depth of 0.5 m; the fluid densities of the upper ($\rho_1$) and lower ($\rho_2$) layers are 996 kg/m$^3$ and 1030
kg/m$^3$, respectively; the ISW amplitude ($a$) is 0.068 m; the location of the centre of the pycnocline ($z_{pyc}$)
is 0.4 m, the pycnocline thickness ($d_{pyc}$) is 0.04 m vertically, and the depths of the upper ($h_1$) and lower



($h_2$) layers are 0.1 m and 0.4 m, respectively. The grid is uniform in the *x*-direction, *y*-direction and *z*-
direction, and the sizes are $\Delta x = 1.5 \times 10^{-2}$ m, $\Delta y = 1.5 \times 10^{-2}$ m and $\Delta z = 2.72 \times 10^{-3}$ m, respectively. The
sponge layer on both sides, whose length is double wave characteristic length, has been checked to
properly dissipate the reflected wave. Slip boundary conditions are applied to the bottom and both sides,
while cyclic boundary conditions are assigned to the inlet and outlet boundaries. The top boundary is a
rigid lid. The boundary conditions related to the density field are no-flux boundary conditions.

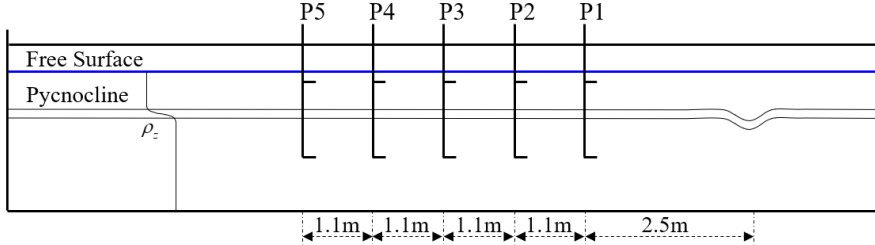


Figure 6: Schematic diagram of probe position (P1–P5) (Hsieh et al. (2014)).

**4.1.2 Comparisons between the numerical and experimental results**

Fig. 7 shows the density contours at three different times from Case Flat_4 in the laboratory

experiment of Hsieh and coworkers, showing the stable evolution of an ISW. The results also show the
realistic evolution of the thickening of the pycnocline after ISW propagation because of convection and
diffusion. At the same time, the propagation of the ISW is stable and unbroken.

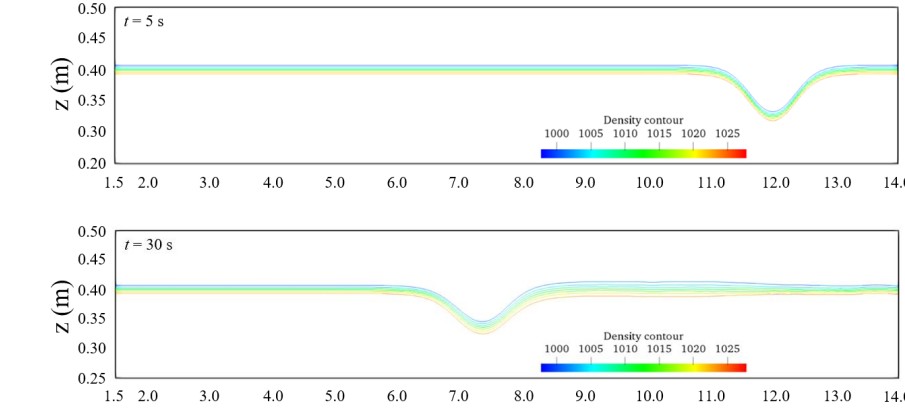



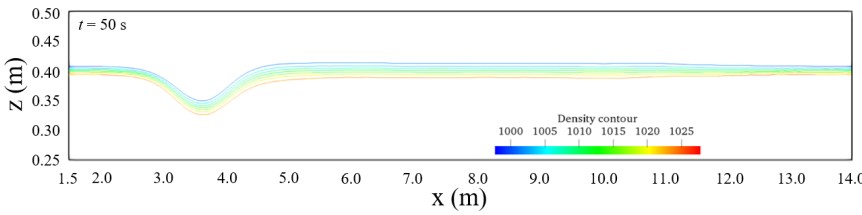


Figure 7: Density contours at different moments.

To further verify the model, the waveform is compared between the numerical simulations and the
experimental measurements, and the measurement point selection is the same as the experimental setting,
as shown in Fig. 6. Fig. 8 shows the comparison results between the waveform simulated by ISWFoam
and the experimental results at probes P1-P5. Fig. 8 shows that the results of the numerical simulations
agree with the experimental results (red circle). Notably, the laboratory wave height at the probe P1
measurement point is greater than the numerical simulation results, and the wave surface of the laboratory
wave is not smooth, which is caused by the wave generation method using the gravity collapse
mechanism in the laboratory. In general, the model developed in this paper can simulate the generation
and evolution of ISWs in continuously stratified fluids.

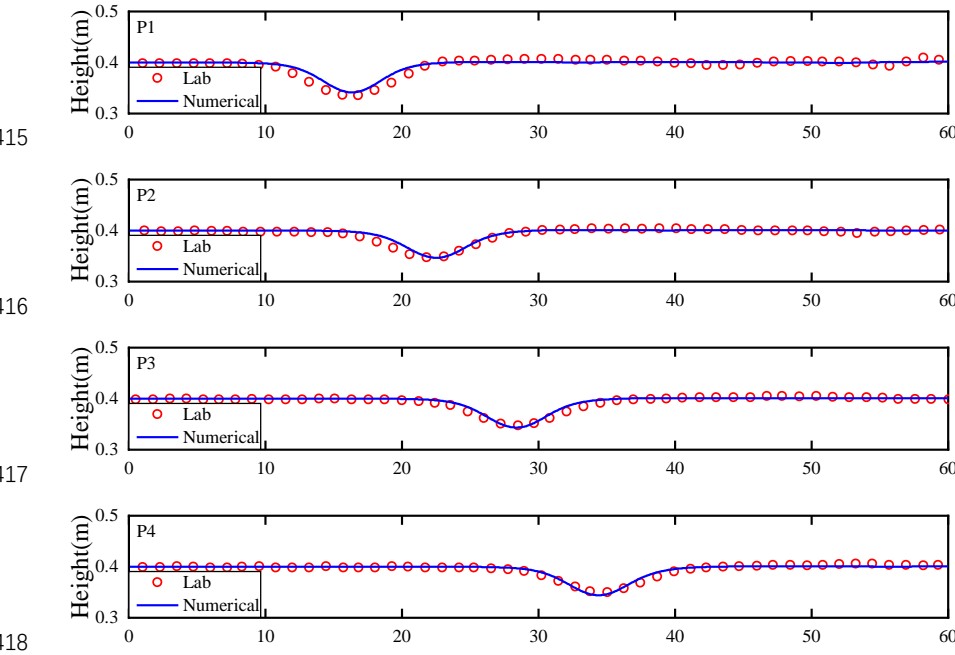









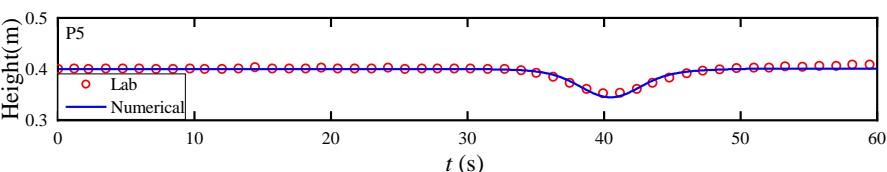


Figure 8: Comparison of the waveform between the experimental results and numerical simulation results at

probes P1-P5.

**4.2 ISW propagating over a submerged triangular ridge**

**4.2.1 Experimental data used**

In this section, the validation of the numerical model is conducted through an ISW propagating over

a submerged triangular ridge with the continuously stratified experiments described in Hsieh et al. (2015).
The laboratory tank is 12 m long and has a stable water depth of 0.5 m, with which the fluid system has
a finite thickness of the pycnocline. The specific experimental parameters used for validation of
ISWFoam include the various depths of the upper ($h_1$) and lower ($h_2$) layers; the fluid density of the upper
($\rho_1$) and lower ($\rho_2$) layers of 996 kg/m$^3$ and 1030 kg/m$^3$, respectively; the ISW amplitude ($\alpha$ = 0.056 m);
the location of the centre of the pycnocline ($z_{pyc}$ = 0.4 m); the thickness of the pycnocline ($d_{pyc}$ = 0.04 m
vertically); the height of the isosceles triangular ridge ($h_s$ = 0.30 m vertically); and the slope angle of the
ridge for $\alpha$ = 45°. The physical dimensions, and ultrasonic probe locations in the experiments of Hsieh
et al. (2015), as shown in Fig. 9, are adopted to establish the numerical computation domain.

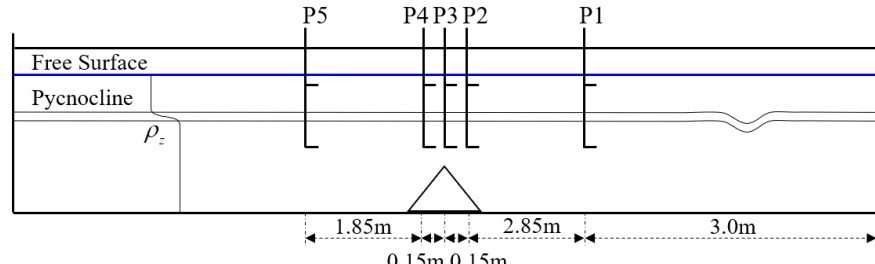


Figure 9: Schematic illustration of the laboratory setup and the locations of the probes (Hsieh et al. (2015)).

**4.2.2 Numerical implementation**

The numerical tank is designed to reproduce the experiment described in Fig. 9. The unstructured

grid and local mesh refinement based on the finite volume method are used to construct the computational
domain and discretize the governing equations. The grid is uniform in the *x*-direction, *y*-direction and *z*-



direction, and the sizes are $\Delta x = 2\times10^{-3}$ m, $\Delta y = 2\times10^{-3}$ m and $\Delta z = 2\times10^{-3}$ m, respectively. The precise
grid described triangular ridge section is $\Delta x = 2.5\times10^{-4}$ m, $\Delta y = 2.5\times10^{-4}$ m and $\Delta z = 2.5\times10^{-4}$ m at the
slope, as shown in Fig. 10. The sponge layer on both sides, whose length is double the wave characteristic
length defined through integration of the wave profile in Section 3 for this case, has been checked to
absorb the reflected wave well. A rigid wall conditions is applied to both sides, while the slip and slip
conditions are assigned to the bottom and the surface of the submerged ridge boundaries, respectively.
The top boundary is a rigid lid. The inlet and outlet boundaries adopt cyclic boundary condition. The
boundary conditions related to the density field are no-flux boundary conditions.

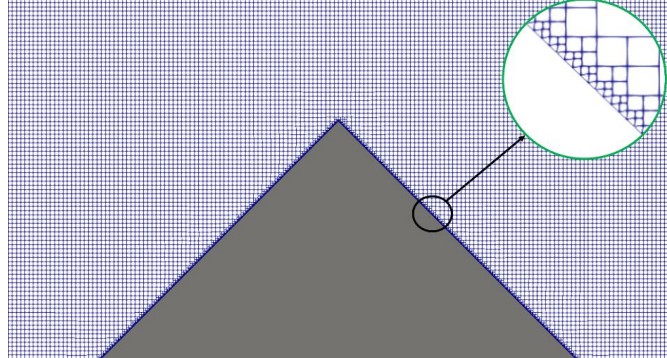

Fig. 10. Schematic of the mesh
**4.2.3 Comparisons between the numerical and experimental results**
Fig. 11 shows the comparison results between the waveform calculated by ISWFoam and the
experimental results at probes P1-P5. In each subplot, the results of the numerical simulations (blue line)
are found to be in good agreement with the experimental results (red circle). From Fig. 11 (a), the
numerical simulation result of the probe P1 measurement after 20 s is different from the experimental
results, which is caused by the different ISW generation methods. For the experimental results, the first
large leading ISW is formed via the gravity collapse mechanism, which is trailed by a train of small-
amplitude mode-one waves that is generated due to shear instabilities. However, the initialization method
used to generate an initial ISW for the numerical simulation in this paper is more stable than the gravity
collapse mechanism, so the rear part of the ISW is relatively flat compared to the experimental results
for probe P1. In Fig. 11, the waveform of the ISW gradually evolves towards a flat waveform due to the
interaction between the ISW and the ridge. In general, the model developed in this paper can simulate
the interaction between ISWs and structures.



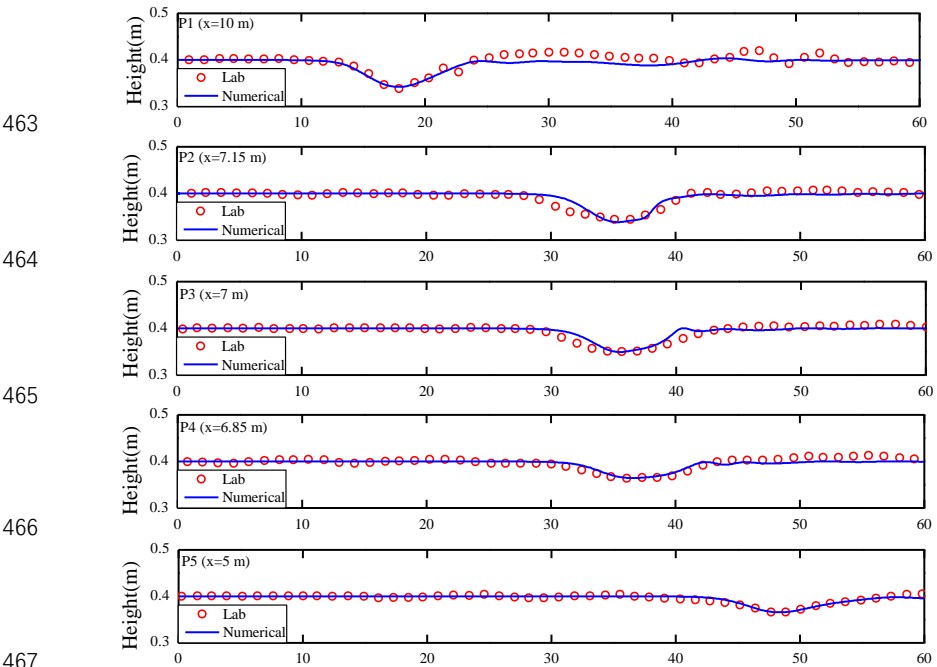






Figure 11: Comparison of the waveform between the experimental results in Hsieh et al. (2015) and numerical

simulation results at probes P1-P5.

**4.3 ISW propagating on a slope**
To verify the ability and accuracy of simulating the ISW breaking of the numerical model, two
continuously stratified laboratory experiments (12 and 15) of Michallet and Ivey. (1999) are chosen for
the simulation in this section. The experimental setup is represented schematically in Fig. 12. We set up
a numerical tank of the experiment of Michallet and Ivey. (1999), which includes a tank is $L$= 4.2 m long,
0.25 m wide and has a water depth of 0.15 m. The layer thickness ratio ($h/H$) varies from 0.60~0.91. A
linear slope $s$ = 0.214 starts at 0.7 m from the right end of the tank for experiments 12 and 15. The grid
is uniform in the $x$-direction, $y$-direction and $z$-direction, and the sizes are $\Delta x$ = 2.5×10$^{-3}$ m, $\Delta y$ = 2.5×10$^{-}$
$^{3}$ m and $\Delta z$ = 1.25×10$^{-3}$ m, respectively. The precise grid describing the slope section is $\Delta x = \Delta y$ = 6.25×10$^{-}$
$^{4}$ m and $\Delta z$ = 3.125×10$^{-4}$ m at the slope.





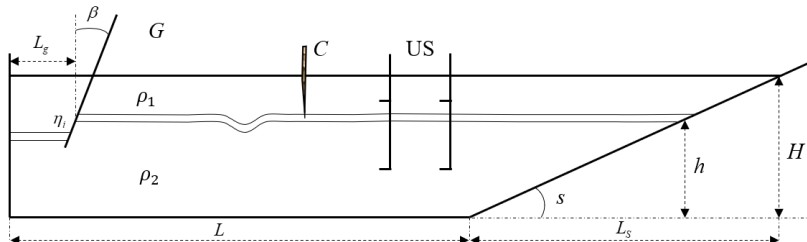


Figyre 12: Schematic diagram of the laboratory setup. "C" and "US" represent the experimental device at probes.

The sponge layer on the left side, whose length is the double wave characteristic length, is checked

to properly dissipate the reflected wave. Slip boundary conditions are applied to the bottom and both
sides, while slip boundary conditions are assigned to the top boundaries. The boundary conditions related
to the density field are no-flux boundary conditions.

The vertical profile of the initial density is given by a hyperbolic tangent function profile

$$\overline{\rho}\left(z\right) = \rho_1 + \frac{\Delta\rho}{2}\left\{1 + \tanh\left[\frac{-\left(z - z_{pyc}\right)}{d_{pyc}}\right]\right\} \tag{44}$$

where $z$ is the vertical position, $\rho_1 = 1\times10^3$ kg/m$^3$ is the base density field, $\Delta\rho$ is the change in the density,
$z_{pyc}$ is the location of the centre of the pycnocline, and $d_{pyc}$ is the thickness of the pycnocline.

**4.3.1 Case one and results**

The first case of model verification is experiment 12 of Michallet and Ivey. (1999) in this section.

The layer thickness ratio ($h/H$) is 0.84, and the density change ($\Delta\rho$) value is 14 kg/m$^3$. Fig. 13 presents
the time series for the interface displacement ($\zeta$) for experiment 12 of Michallet and Ivey. (1999). The
results indicate reasonably good agreement between the time series of the simulated interface
displacement and that of the laboratory results. The first trough centred around $t = 25$ s represents the
incident ISW propagating the probe 99.8 cm away from the start of the slope. The second trough centred
at approximately $t = 87$ s represents the reflected ISW at the generation side, which has a smaller
amplitude and a longer wavelength than the incident ISW as the energy in the wave decreases. As shown
in Fig. 13, the smooth waveform of the incident ISW of the numerical simulation indicates that the
initialization method of wave generation in this paper is more stable than the experiment.

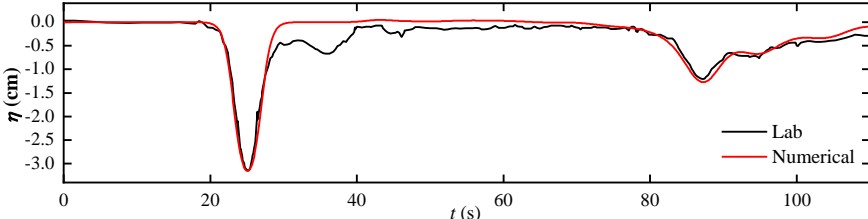

Figure 13: Time series of the interface displacement. The probe was 99.8 cm away from the start of the slope.
Fig. 14 shows a comparison of ISWFoam results and the experimental observations of the velocity
field associated with the ISW run-up process along the slope. The model effectively reproduces
laboratory tests, such as the intensity and direction of the velocity field, the location of the vortices, and
the occurrence of boundary-layer separation beneath the ISW. Therefore, the model developed in this
paper can reflect the ISW breaking phenomenon during the propagation of ISWs along the slope.

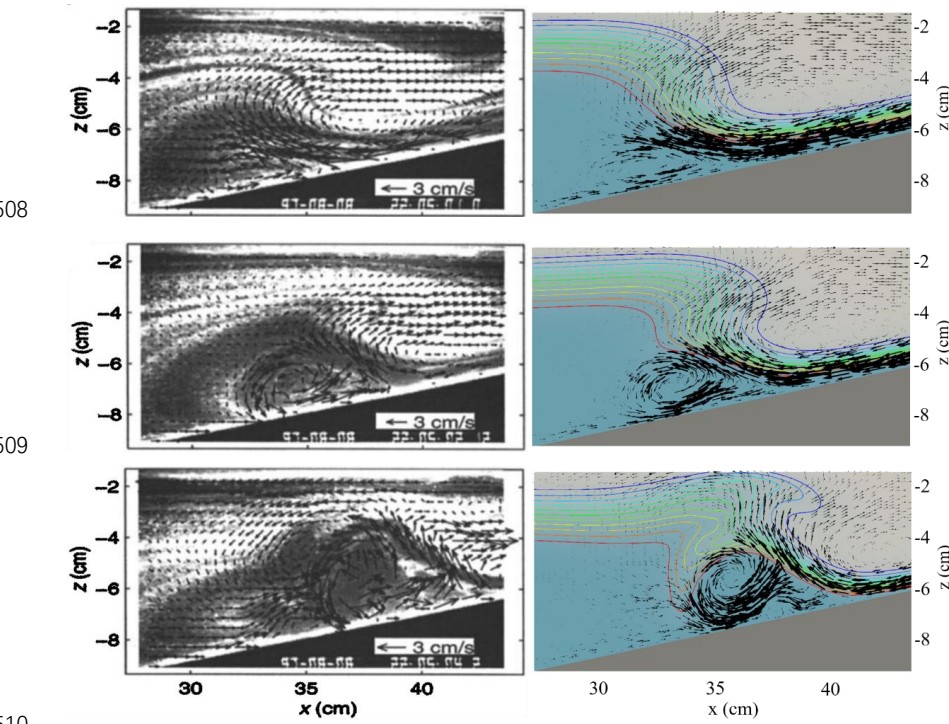

Figure 14: Comparison of the velocity field between the experimental observation results in Michallet and Ivey.
(1999) (left) and numerical simulation results (right).

**4.3.2 Case two and results**
Another laboratory experiment that more clearly shows the ISW breaking phenomenon from





Experiment 15 of Michallet and Ivey. (1999) is used to verify the numerical model presented in this paper,
and the corresponding numerical case is set corresponding to it. The layer thickness ratio ($h/H$) is 0.77,
and the density change ($\Delta\rho$) value is 47 kg/m$^3$. The wave amplitude and phase velocity at the slope
calculated by ISWFoam are $2.71\times10^{-3}$ m and $10.83\times10^{-1}$ m/s, which fit well with the experimental results
of $2.7\times10^{-3}$ m and $10.8\times10^{-1}$ m/s.

Fig. 15 shows the results of the numerical simulations of the ISWs propagating along the slope and

wave breaking using ISWFoam. As the ISW propagates to the slope, according to the conservation of
mass, the upper fluid forward and the downward velocity of the lower fluid increasing along the slope
results in the formation of a thin boundary layer, as shown in Fig. 15($a$), ($b$), and ($c$). At the same time,
the amplitude of the ISW increases, and the rear of the ISW gradually becomes very steep but does not
overturn. With the development of the ISW, the rear waveform of the ISW cannot maintain its stability
and overturns forward, resulting in wave breaking, as shown in Fig. 15($d$). After wave breaking occurs,
the denser lower layer flow accelerates into the less dense upper layer flow, forming a mixture region, as
shown in Fig. 15($e$). After the lower layer flow is drawn downward from beneath the ISW, a mixing
region comprised of vortices is pushed upwards along the slope while the leading waveform is reflected,
as shown in Fig. 15($f$), ($g$), and ($h$). Fig. 15($i$), ($j$) shows the falling process of ISWs. From the perspective
of the entire process of wave breaking, the steepening of the rear waveform in this case is the main reason
for wave breaking.

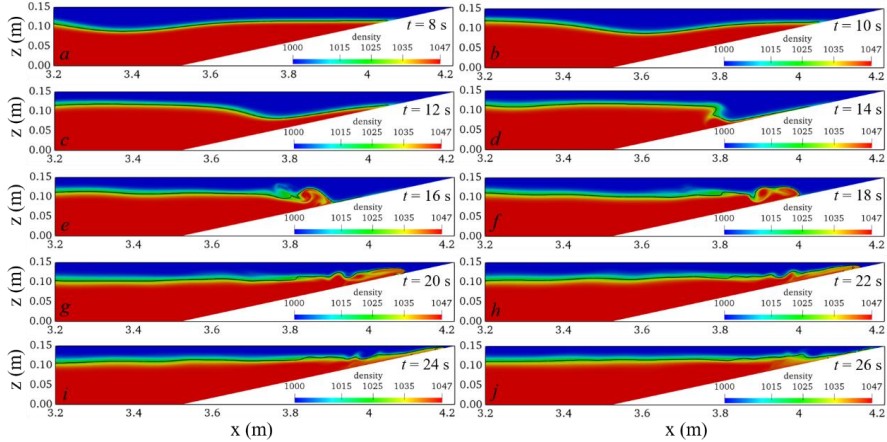


Figure 15: Temporal and spatial variations in the ISWs breaking calculated using ISWFoam (the black line

represents the waveform).

For comparison with the flow visualization image of the experiment, a specified thickness of the



pycnocline is presented, and the pycnocline ranges from 1003 kg/m³ to 1045 kg/m³ with dark colours as
shown in Fig. 16.
Fig. 16 compares the ISWFoam results and the experimental results of Michallet and Ivey. (1999)
before, during, and after ISW breaking. The results indicate that some main features of the laboratory
tests are reasonably well reproduced by ISWFoam, such as the profile of ISW, the location of the wave
breaking point, ISW arrival time, and spatial and temporal changes in the mixture region. Therefore, the
model developed in this paper can accurately simulate the ISW breaking phenomenon during the
propagation of ISWs along the slope.

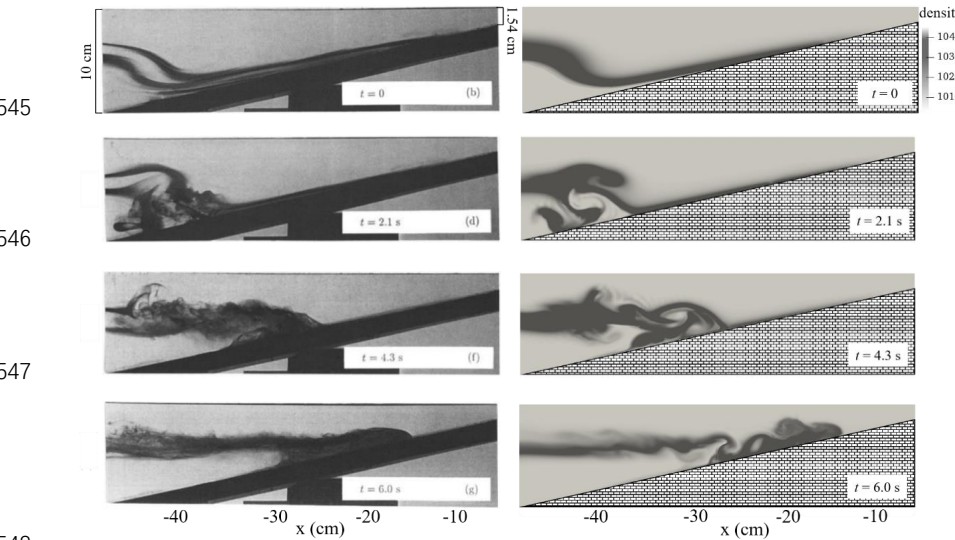

Figure 16: Comparison of the density fields between the experimental observation results in Michallet and Ivey.
(1999) (left) and the numerical simulation results (right).

**4.4 Coriolis force analysis**

Notably, the previous verifications are all laboratory-scale verifications, thus, it is difficult to clearly
see the influence of the Coriolis force on ISWs. To verify the correctness of the Coriolis force added in
the code, we set up a 3D numerical tank, which includes a tank that is 15 km long, 1 km wide ($y$ direction
from -500 m to 500 m) and has a water depth of 300 m. The depths of the upper ($h_1$) and lower ($h_2$) layers
are 50 m and 250 m, respectively, the densities of the upper and lower layers are 1022 kg/m³ and 1028
kg/m³, respectively, the location of the centre of the pycnocline ($z_{pyc}$) is 250 m vertically, and the
pycnocline thickness ($d_{pyc}$) is 5 m vertically, the ISW amplitude ($a$) is 40 m. The grid gradually changes



from $\Delta x = 100$ m to $\Delta x = 10$ m in the $x$-direction, the grids in the $y$-direction are uniform with a constant
cell width of $\Delta y = 10$ m, and the grids in the $z$-direction are non-uniform, with a minimum cell height of
$\Delta z = 2.5$ m near the interface of the ISW. To see the influence of the Coriolis force more clearly, the
Coriolis force parameter ($\Omega$) is expanded 5 times, to $3.65 \times 10^{-4}\,\text{s}^{-1}$. The sponge layer on both sides, whose
length is the double wave characteristic length, has been checked to properly dissipate the reflected wave.
Slip boundary conditions are applied to the bottom and both sides, while cyclic boundary conditions are
assigned to the inlet and outlet boundaries. The top boundary is a rigid lid. The boundary conditions
related to the density field are no-flux boundary conditions.

Figs 17 and 18 show the two-dimensional (2D) and three-dimensional (3D) waveform diagrams

when the ISW propagates for 5000s. The figures show that under the influence of the Coriolis force,
asymmetry is caused in the front and back wave surfaces, and the height difference is approximately 6
m. Since this summary only verifies the correctness of the Coriolis force implanted in the code, there is
no further quantitative analysis of the influence of the Coriolis force on ISWs. In general, ISWFoam can
effectively reflect the influence of the Coriolis force. At the same time, the influence of the Coriolis force
cannot be ignored when performing large-scale simulations.

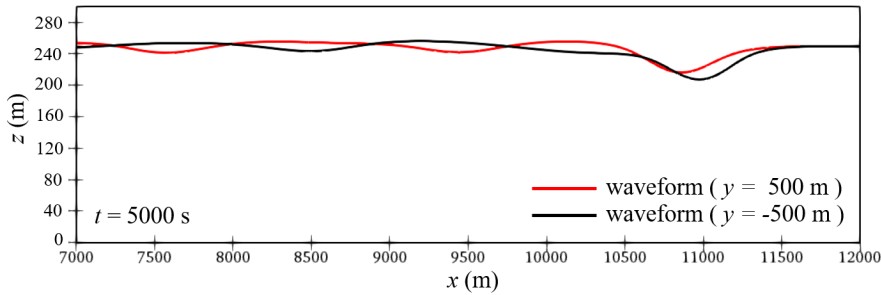


Figure 17: Schematic of the waveforms on the front ($y = -500$) and back ($y = 500$).



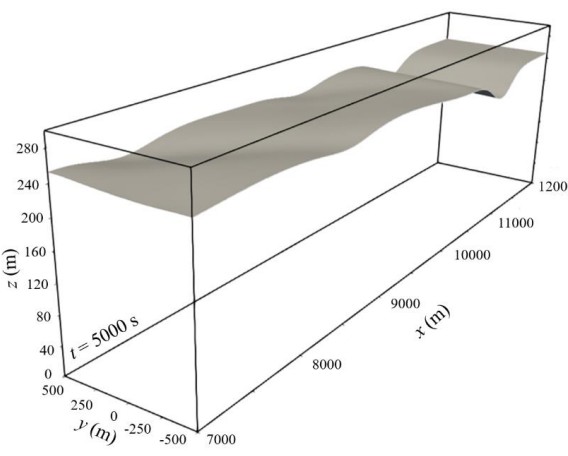

Figure 18: Schematic of the three-dimensional waveform.

**4.5 ISW propagating over a 3D Gaussian ridge**

The ISWFoam model with unstructured grids and local mesh refinement can accurately simulate

the interaction between ISWs and complex structures and topography. However, the terrain in the

previous cases is relatively simple, and it is difficult to see such characteristics. Therefore, we designed

a case of an ISW propagating over a 3D Gaussian ridge. The 3D Gaussian ridge is obtained by rotating

a 2D Gaussian ridge

$$z = ae^{-(x/l)^2} \tag{45}$$

where $a$ is the ridge amplitude, and $l$ is the standard deviation.

With $a$ = 5 m and $l$ = 0.5, we can obtain a 2D Gaussian ridge with a height of 5 m and a bottom width of

2 m. Subsequently, the 3D Gaussian ridge can be obtained after a vertical rotation of 180 degrees.

We set up a 3D numerical tank, which includes a tank that is 150 m long, 20 m wide ($y$-direction

from -10 m to 10 m) and has a water depth of 6 m. The depths of the upper ($h_1$) and lower ($h_2$) layers are

1 m and 5 m, respectively, the densities of the upper and lower layers are 1000 kg/m³ and 1030 kg/m³,

respectively, the location of the centre of the pycnocline ($z_{pyc}$) is 5 m, and the pycnocline thickness ($d_{pyc}$)

is 0.1 m vertically, the ISW amplitude ($a$) is 1 m. The Gaussian ridge is located at 80 m horizontally. The

grid is gradually changed from $\Delta x$ = 5 m to $\Delta x$ = 0.2 m in the $x$-direction, the grids in the $y$-direction are

uniform with a constant cell width of $\Delta y$ = 0.2 m, and the grids in the $z$-direction are non-uniform, with

a minimum cell height of $\Delta z$ = 0.05 m near the interface of the ISW. The precise grid described the 3D



Gaussian ridge section as $\Delta x = 1.25 \times 10^{-2}$ m, $\Delta y = 1.25 \times 10^{-2}$ m and $\Delta z = 6.25 \times 10^{-3}$ m, as shown in Fig 19.
The sponge layer on both sides, whose length is the double wave characteristic length, has been checked
to properly dissipate the reflected wave. Slip boundary conditions are applied to the bottom and both
sides, while cyclic boundary conditions are assigned to the inlet and outlet boundaries. The top boundary
is a rigid lid. The boundary conditions related to the density field are no-flux boundary conditions.
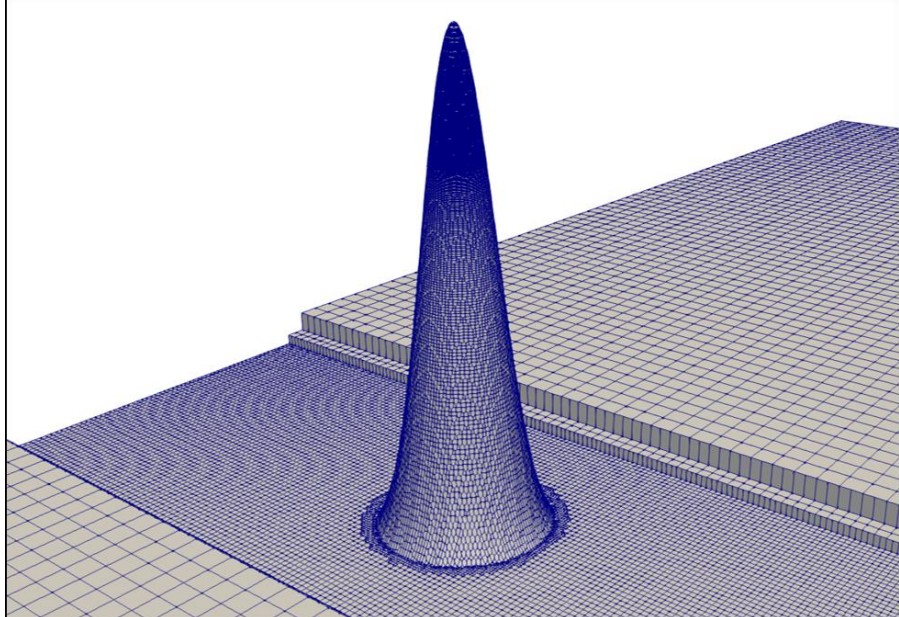
Figure 19: Schematic of the local refinement of the grid.

Fig. 20. shows the temporal and spatial variations in the ISWs propagating over a 3D Gaussian

ridge. The ISW reaches the Gaussian ridge, causing the wave surface in front of the ridge to decrease,
and the wave surface behind the ridge to climb up the ridge, as shown in Fig. 20($a$). Due to being
obstructed by the Gaussian ridge, flow around a ridge and wave surface uplift are generated on both sides
of the Gaussian ridge (perpendicular to the direction of wave propagation), as shown in Fig. 20($b$). As
the ISW propagated over the Gaussian ridge, the wave surface climbed along the ridge, and at the same
time, low velocity was generated behind the ridge, as shown in Fig. 20($c$). Since the top of the ridge is in
the pycnocline, there will be a low velocity area behind the ridge for a period of time after the ISW passes,
as shown in Fig. 20($d$). In general, the ISWFoam model with unstructured grids and local mesh
refinement can simulate the interaction between ISWs and complex structures and topography.

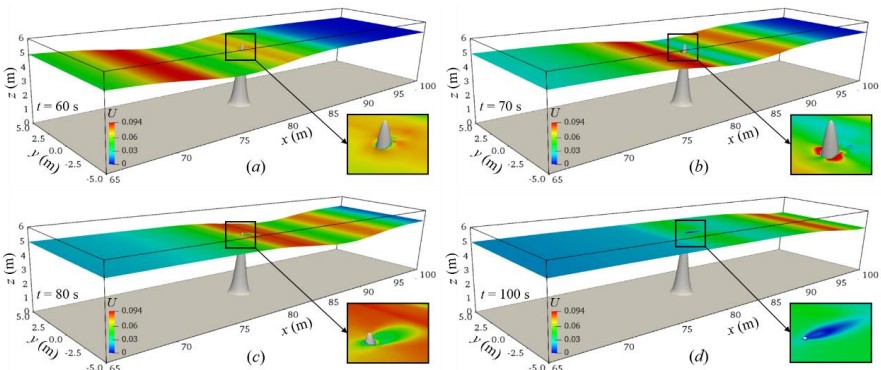

Figure 20: Temporal and spatial variation in the ISWs propagating over a 3D Gaussian ridge.

**5. Conclusions**

In this paper, a numerical model referred to as ISWFoam with a modified $k$-$\omega$ *SST* model, established by combining the density transport equation with a fully three-dimensional (3D) Navier-Stokes equation, is developed to simulate ISWs in continuously stratified, incompressible, viscous fluids based on the finite volume method with unstructured grids and local mesh refinement of OpenFOAM. ISWFoam provides two initial wave generation methods to generate an ISW in continuously stratified fluids, including solving the weakly nonlinear models of the eKdV equation and the fully nonlinear models of the DJL equation. The verification process presents several applications, such as ISWs propagating on flat bottoms including laboratory scale and actual ocean scale, and ISWs over a submerged triangular ridge, a Gaussian ridge and slopes. The following conclusions were obtained as a result of this study.

ISWFoam using the finite volume method with unstructured grids and local mesh refinement can accurately simulate the generation and evolution of ISWs, the ISW breaking phenomenon and the interaction between ISWs and complex structures and topography. The method of initializing the ISW using weakly nonlinear eKdV equation models requires a period of movement before the jump of the velocity field develops into a field with continuous changes in velocity. The DJL equation wave generation method that considers the vertical velocity and the horizontal velocity along the vertical gradient is better than the eKdV equation wave generation method that only provides the horizontal average velocity. Using ISWFoam to simulate an ISW with infinite wave length, the metric for the appropriate mesh size is given as follows: the dimensions of the horizontal grid are one-one hundred and





fiftieth of the characteristic length, while the vertical grid takes one-twenty fifth of the ISW amplitude.

**Computer code availability**
The ISWFoam code developed in this article can be downloaded for free from
https://github.com/Mr-trekking/ISW.git.
**Author contributions**
QZ and JL jointly developed this numerical method to calculate internal solitary waves in
continuously stratified fluids. JL developed the code. TC performed the computations. QZ and JL
jointly analysed the calculation results and wrote the paper together.
**Competing interest**
The authors of this paper declare that they have no conflicts of interest.
**Financial support**
This work is supported by grants from the National Key Research and Development Program of
China (2017YFC1404200), and the National Natural Science Foundation of China (51509183).

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
