# Peer review of "ISWFoam: A numerical model for internal solitary wave 1"

_Geoscientific Model Development, 2021_

## Author Comment (AC2)

**Answers to referee #1**

**General Comments**

Based on the open source code OpenFOAM the authors developed some modules to enable the Foam model for simulating internal solitary waves (ISWs) in continuously stratified fluids. To resolve the case of continuously stratified fluids the k-ω SST turbulence model was modified accordingly for the variable density field. The authors proved the model performance via a series of convincing model verifications. This paper also provides two options to generate an ISW in continuously stratified fluids, the fully nonlinear models of the Dubreil-Jacotin-Long (DJL) equation method and the extended Korteweg–de Vries (eKdV) equation method. This work is a good contribution to OpenFOAM model package.

However, this model development is limited to laboratory scale ISW modelling, instead of ISW modelling for realistic oceans. k-ω SST turbulence model resolve only density case instead of temperature and salinity case, thus, this model is not applicable for real ocean waves' modelling. But a nice job for laboratory scale modelling.

**Specific Comments**

Q1: For realistic ocean modelling, the important item for a ISW model is to specific ISW or generate ISW by open boundary conditions. Since no open boundary codes are developed and tested for this ISWFoam, I would suggest the authors emphasize not the modelling scale for actual ocean scale. You also have to consider temperature and salinity by applying other turbulence model for real ocean scale. A realistic density field should also be considered for model validation.

Answer: Thanks for the suggestions. It is worth noting that ISWFoam does not consider the generation process of ISWs, but focuses on the propagation and evolution of ISWs that have already been generated, and the interaction between ISWs and complex structures and topography on actual scales. In the revised version, we have added Section 5 to illustrate the application of ISWFoam at the actual ocean scale. In Section 5, the evolution of the vortex structure, the waveform inversion and breaking phenomenon of ISWs are well indicated, and the propagation and evolution of the wave train generated by waveform inversion is also accurately described through ISWFoam simulation.

Q2: Section 4.4 of 'Coriolis force analysis' should be reconsidered. The authors designed this numerical experiment just for proving the Coriolis' effect. However, this experiment should have been carefully designed. Such model settings using a 12km-long tank is not convincing for Coriolis' effect. I would suggest the authors repeat this modelling experiment using real laboratory scale (maximum 12 meter, like the ISW tank in France), or just remove Section 4.4.

Answer: Thanks for the suggestions. We have deleted section 4.4.

---

## Author Comment (AC3)

**Answers to referee #2**

**General Comments**

This article uses the ISWFoam model developed based on OpenFOAM to realize the simulation study of the generation, propagation and evolution of the internal solitary wave (ISW). By comparing with the experimental data, the authors point out that the ISWFoam model with unstructured grids and local mesh refinement can accurately simulate the generation and evolution of ISWs, the ISW breaking phenomenon and the interaction between ISWs and complex structures and topography. Due to the interpretation of the article and the code, some questions need to be solved. Compared with the original code in OpenFOAM, ISWFoam does not reflect its own characteristics and innovation. The ISWFoam built in this paper is an integration of the OpenFOAM base tools, rather than a new developed code. Compared to the existing works on the ISW simulated by OpenFOAM, ISWFoam does not show its advantages and comprehensiveness. Based on the above reasons, I suggest to reject this manuscript.

**Specific Comments**

Q1: 1. The manuscript mentions that "the wave generation method is essential for a two-layer system" on Page3 line 85, as described in the manuscript and code, the ISWFoam generates the ISW by the horizontal velocity derived from ISW theory. The corresponding code is in 'setUFields.C'. The initial density distribution in the flow field is established by the ISW theory with the hyperbolic tangent function profile. The corresponding code is in 'setRhoFields.C'. The mere comparison of the DJL equation and the eKdV equation does not show that the ISW generation method used in this manuscript is excellent. More equations including KdV, mKdV, MCC et al., should be examined in the ISWFoam. The article does not do enough work on ISW generation. In addition, the initial flow fields can be set using 'setFields' in OpenFOAM and 'funkySetFields' in swak4foam.

Answer: The complete sentence in the paper is "*Though recent work by Ding et al (2020) and Li et al (2021) considered continuous stratification in density, the wave generation method is essential for a two-layer system*". The objective of this sentence is to show that Ding et al (2020) and Li et al (2021) considered continuous density stratification in their model, but their wave generation theories are still strongly stratified (a two-layer system), does not consider continuous stratification in density, which is inconsistent with the actual situation. However, ISWFoam not only considers continuous density stratification in the solution process, but also considers continuous density stratification in the wave generation theory (the ISW generated by the fully nonlinear models of the Dubreil-Jacotin-Long (DJL) equation). Section 2.3.1 also gives the difference in wave generation with and without considering continuous stratification in density. The results show that the wave generation theory

considering continuous density stratification is more reasonable. It is worth noting that ISWFoam is not a two-layer system (such as interFoam), which can also be seen from the code. To express the idea more clearly, we have changed the sentence as follows. "Though recent work by Ding et al (2020) and Li et al (2021) considered continuous stratification in density, their wave generation theories does not consider continuous stratification in density."

The wave generation of the numerical cases in this paper adopts the method of initializing the field with the fully nonlinear models of the DJL equation. The purpose of comparing the DJL and eKdV equations is just to highlight that the DJL equation is more reasonable (although it can be obtained from the equation itself), so other weakly nonlinear theories such as KdV, mKdV, MCC et al are not discussed in depth.

Without modifying the code, **neither** OpenFoam's original functions setFields and funkySetFields in swak4foam can solve the initial field of ISWs.

Q2: According to the introduction of the governing equations in the article and the code, by taking the variation in density into account, 'interFoam' enables a simulation study of ISW, and the value of the authors' work is not reflected. The section 2, which describes the model and the various methods, also comes with OpenFOAM and can be found in the user manual. The article should describe the characteristics of ISWFoam and how it differs from the original program.

Answer: At present, the official version of OpenFOAM® does not have a solver or boundary conditions for solving the ISW in continuously stratified fluids. The two-layer system model interFoam in OpenFOAM is strictly incompressible, and the density of the water is a constant value, and continuous density stratification of the water cannot be considered. In order to solve the internal solitary waves in the real ocean environment, a new solver (ISWFoam) was developed by independent programming to simulate internal solitary waves in continuously stratified, incompressible, viscous fluids based on a fully three-dimensional (3D) Navier-Stokes equation using the open source code OpenFOAM. The turbulence model has also been modified accordingly to the variable density field.

Q3: It has been explained in the Introduction that ISW research has already been implemented using OpenFOAM, what are the differences or advantages of ISWFoam from those existing codes?

Answer: At present, the official version of OpenFOAM® does not have a solver or boundary conditions for solving the ISW in continuously stratified fluids. Incompressible fluid solver in OpenFOAM is strictly incompressible, and the density of the water is a constant value, and continuous density stratification of the water cannot be considered. However, ISWFoam not only

considers continuous density stratification in the solution process, but also considers continuous density stratification in the wave generation theory (the ISW generated by the fully nonlinear models of the Dubreil-Jacotin-Long (DJL) equation).

In the introduction, it is introduced that some researchers simulate ISWs by modifying the OpenFOAM® code, most of these studies are based on a two-fluid system (for example interFoam) without continuous density stratification in the solution process, such as Meng and Zhang (2016) and Li et al (2017). Though recent work by Ding et al (2020) and Li et al (2021) considered continuous stratification in density, their wave generation theories does not consider continuous stratification in density. ISWFoam not only considers continuous density stratification in the solution process, but also considers continuous density stratification in the wave generation theory. At the same time, the turbulence model has also been modified accordingly to take account of .

At the same time, the turbulence model has been modified accordingly to take account of continuous stratification in density.

Q4: The meaning of the star icon in Figs. 4 and 5 should be indicated.

Answer: Thanks for the suggestions. We have revised and removed the star icon.

Q5: If a rigid lid is used for the top boundary, then the free surface should not be labelled in Figs. 6 and 7.

Answer: Thanks for the suggestions. Fig. 6 is the experimental diagram, which we draw in accordance with the experimental layout diagram given by Hsieh et al. (2014). And Fig. 7 is the numerical simulation result, we did not mark the free surface.

---

## Author Response (AR3)

**Answers to referee #1**

**General Comments**

Based on the open source code OpenFOAM the authors developed some modules to enable the Foam model for simulating internal solitary waves (ISWs) in continuously stratified fluids. To resolve the case of continuously stratified fluids the k-ω SST turbulence model was modified accordingly for the variable density field. The authors proved the model performance via a series of convincing model verifications. This paper also provides two options to generate an ISW in continuously stratified fluids, the fully nonlinear models of the Dubreil-Jacotin-Long (DJL) equation method and the extended Korteweg–de Vries (eKdV) equation method. This work is a good contribution to OpenFOAM model package.

However, this model development is limited to laboratory scale ISW modelling, instead of ISW modelling for realistic oceans. k-ω SST turbulence model resolve only density case instead of temperature and salinity case, thus, this model is not applicable for real ocean waves' modelling. But a nice job for laboratory scale modelling.

**Specific Comments**

**Q1**: For realistic ocean modelling, the important item for a ISW model is to specific ISW or generate ISW by open boundary conditions. Since no open boundary codes are developed and tested for this ISWFoam, I would suggest the authors emphasize not the modelling scale for actual ocean scale. You also have to consider temperature and salinity by applying other turbulence model for real ocean scale. A realistic density field should also be considered for model validation.

Answer: Thanks for the suggestions. It is worth noting that ISWFoam does not consider the generation process of ISWs, but focuses on the propagation and evolution of ISWs that have already been generated, and the interaction between ISWs and complex structures and topography on actual scales. In the revised version, we have added Section 5 to illustrate the application of ISWFoam at the actual ocean scale. In Section 5, the evolution of the vortex structure, the waveform inversion and breaking phenomenon of ISWs are well indicated, and the propagation and evolution of the wave train generated by waveform inversion is also accurately described through ISWFoam simulation.

**Q2**: Section 4.4 of 'Coriolis force analysis' should be reconsidered. The authors designed this numerical experiment just for proving the Coriolis' effect. However, this experiment should have been carefully designed. Such model settings using a 12km-long tank is not convincing for Coriolis' effect. I would suggest the authors repeat this modelling experiment using real laboratory scale (maximum 12 meter, like the ISW tank in France), or just remove Section 4.4.

Answer: Thanks for the suggestions. We have deleted section 4.4.

**Answers to referee #2**

General Comments

This article uses the ISWFoam model developed based on OpenFOAM to realize the simulation study of the generation, propagation and evolution of the internal solitary wave (ISW). By comparing with the experimental data, the authors point out that the ISWFoam model with unstructured grids and local mesh refinement can accurately simulate the generation and evolution of ISWs, the ISW breaking phenomenon and the interaction between ISWs and complex structures and topography. Due to the interpretation of the article and the code, some questions need to be solved. Compared with the original code in OpenFOAM, ISWFoam does not reflect its own characteristics and innovation. The ISWFoam built in this paper is an integration of the OpenFOAM base tools, rather than a new developed code. Compared to the existing works on the ISW simulated by OpenFOAM, ISWFoam does not show its advantages and comprehensiveness. Based on the above reasons, I suggest to reject this manuscript.

**Specific Comments**

Q1: 1. The manuscript mentions that "the wave generation method is essential for a two-layer system" on Page3 line 85, as described in the manuscript and code, the ISWFoam generates the ISW by the horizontal velocity derived from ISW theory. The corresponding code is in 'setUFields.C'. The initial density distribution in the flow field is established by the ISW theory with the hyperbolic tangent function profile. The corresponding code is in 'setRhoFields.C'. The mere comparison of the DJL equation and the eKdV equation does not show that the ISW generation method used in this manuscript is excellent. More equations including KdV, mKdV, MCC et al., should be examined in the ISWFoam. The article does not do enough work on ISW generation. In addition, the initial flow fields can be set using 'setFields' in OpenFOAM and 'funkySetFields' in swak4foam.

Answer: The complete sentence in the paper is "*Though recent work by Ding et al (2020) and Li et al (2021) considered continuous stratification in density, the wave generation method is essential for a two-layer system*". The objective of this sentence is to show that Ding et al (2020) and Li et al (2021) considered continuous density stratification in their model, but their wave generation theories are still strongly stratified (a two-layer system), does not consider continuous stratification in density, which is inconsistent with the actual situation. However, ISWFoam not only considers continuous density stratification in the solution process, but also considers continuous density stratification in the wave generation theory (the ISW generated by the fully nonlinear models of the Dubreil-Jacotin-Long (DJL) equation). Section 2.3.1 also gives the difference in wave generation with and without considering continuous stratification in density. The results show that the wave generation theory considering continuous density stratification is more reasonable. It is worth noting that ISWFoam

is not a two-layer system (such as interFoam), which can also be seen from the code. To express the idea more clearly, we have changed the sentence as follows. "Though recent work by Ding et al (2020) and Li et al (2021) considered continuous stratification in density, their wave generation theories does not consider continuous stratification in density."

The wave generation of the numerical cases in this paper adopts the method of initializing the field with the fully nonlinear models of the DJL equation. The purpose of comparing the DJL and eKdV equations is just to highlight that the DJL equation is more reasonable (although it can be obtained from the equation itself), so other weakly nonlinear theories such as KdV, mKdV, MCC et al are not discussed in depth.

Without modifying the code, **neither** OpenFoam's original functions setFields and funkySetFields in swak4foam can solve the initial field of ISWs.

Q2: According to the introduction of the governing equations in the article and the code, by taking the variation in density into account, 'interFoam' enables a simulation study of ISW, and the value of the authors' work is not reflected. The section 2, which describes the model and the various methods, also comes with OpenFOAM and can be found in the user manual. The article should describe the characteristics of ISWFoam and how it differs from the original program.

Answer: At present, the official version of OpenFOAM® does not have a solver or boundary conditions for solving the ISW in continuously stratified fluids. The two-layer system model interFoam in OpenFOAM is strictly incompressible, and the density of the water is a constant value, and continuous density stratification of the water cannot be considered. In order to solve the internal solitary waves in the real ocean environment, a new solver (ISWFoam) was developed by independent programming to simulate internal solitary waves in continuously stratified, incompressible, viscous fluids based on a fully three-dimensional (3D) Navier-Stokes equation using the open source code OpenFOAM. The turbulence model has also been modified accordingly to the variable density field.

Q3: It has been explained in the Introduction that ISW research has already been implemented using OpenFOAM, what are the differences or advantages of ISWFoam from those existing codes?

Answer: At present, the official version of OpenFOAM® does not have a solver or boundary conditions for solving the ISW in continuously stratified fluids. Incompressible fluid solver in OpenFOAM is strictly incompressible, and the density of the water is a constant value, and continuous density stratification of the water cannot be considered. However, ISWFoam not only considers continuous density stratification in the solution process, but also considers continuous

density stratification in the wave generation theory (the ISW generated by the fully nonlinear models of the Dubreil-Jacotin-Long (DJL) equation).

In the introduction, it is introduced that some researchers simulate ISWs by modifying the OpenFOAM® code, most of these studies are based on a two-fluid system (for example interFoam) without continuous density stratification in the solution process, such as Meng and Zhang (2016) and Li et al (2017). Though recent work by Ding et al (2020) and Li et al (2021) considered continuous stratification in density, their wave generation theories does not consider continuous stratification in density. ISWFoam not only considers continuous density stratification in the solution process, but also considers continuous density stratification in the wave generation theory. At the same time, the turbulence model has also been modified accordingly to take account of .

At the same time, the turbulence model has been modified accordingly to take account of continuous stratification in density.

Q4: The meaning of the star icon in Figs. 4 and 5 should be indicated.

Answer: Thanks for the suggestions. We have revised and removed the star icon.

Q5: If a rigid lid is used for the top boundary, then the free surface should not be labelled in Figs. 6 and 7.

Answer: Thanks for the suggestions. Fig. 6 is the experimental diagram, which we draw in accordance with the experimental layout diagram given by Hsieh et al. (2014). And Fig. 7 is the numerical simulation result, we did not mark the free surface.

**Answers to editor**

Comments to the author

After reviewing your revised manuscript, I have decided that it is suitable for publication pending the following minor and technical corrections.

**Specific Comments**

**Q1:** Line 104: Please define "rigid lid hypothesis" in terms of boundary conditions employed.

Answer: Thanks for the suggestion. In the revised version, we have added the corresponding definition, the specific revisions are as follows (L118-L120):

The upper boundary ($z = H$, with $H$ the depth of computation domain) is treated as a rigid lid, the kinematic boundary conditions for this boundary are given by

$$u_k(x, y, H, t)=0 \tag{5}$$

**Q2:** Eq. (2): e_3 is a vector and should be bolded. Please fix later occurrences.

Answer: Thanks for the suggestion. In the revised version, we have revised it accordingly (L107 and L115).

**Q3:** Line 122: Please define SST as "Shear Stress Transport".

Answer: Thanks for the suggestion. In the revised version, we have revised it accordingly (L125).

**Q4:** Eq. (11) and (12): Please define Delta t as the time-step.

Answer: Thanks for the suggestion. In the revised version, we have revised it accordingly (L154).

**Q5:** Line 154: "initial moment" should be "current time".

Answer: Thanks for the suggestion. In the revised version, we have revised it accordingly (L158).

**Q6:** I guess Eqns. (33)-(36) represent wave speeds. Please briefly defined the physical significance of the quantities.

Answer: Thanks for the suggestions. In the revised version, we have added a concise definition to Eqns. (33)-(36), the specific revisions are as follows (L260-L262):

where $\zeta$ is the isopycnal vertical displacement; $c_0$ is the linear phase speed; the coefficients $c_1, c_2$ and $c_3$ are functions of the steady background stratification and shear through the linear eigenmode (vertical structure function) of interest (Helfrich and Melville, 2006)

**Q7:** 7. Line 608: Replace "center" with "centre".

Answer: Thanks for the suggestion. In the revised version, we have revised it accordingly (L614).